# Detection of large-scale cloud microphysical changes within a major shipping corridor after implementation of the IMO 2020 fuel sulfur regulations

Michael S. Diamond

Department of Earth, Ocean and Atmospheric Science, Florida State University, Tallahassee, FL, 32306, USA

*Correspondence to*: Michael S. Diamond (msdiamond@fsu.edu)

**Abstract.** New regulations from the International Maritime Organization (IMO) limiting sulfur emissions from the shipping industry are expected to have large benefits in terms of public health but may come with an undesired side effect: acceleration of global warming as the climate-cooling effects of ship pollution on marine clouds are diminished. Previous work has found a substantial decrease in the detection of ship tracks in clouds after the IMO 2020 regulations went into effect but changes in large-scale cloud properties have been more equivocal. Using a statistical technique that estimates counterfactual fields of what large-scale cloud and radiative properties within an isolated shipping corridor in the southeastern Atlantic would have been in the absence of shipping, we confidently detect a reduction in the magnitude of cloud droplet effective radius decreases within the shipping corridor and find evidence for a reduction in the magnitude of cloud brightening as well. The instantaneous radiative forcing due to aerosol–cloud interactions from the IMO 2020 regulations is estimated as $O(1 \text{ W m}^{-2})$ within the shipping corridor, lending credence to global estimates of $O(0.1 \text{ W m}^{-2})$. In addition to their geophysical significance, our results also provide independent evidence for general compliance with the IMO 2020 regulations.

## 1 Introduction and approach

Since 1 January 2020, International Maritime Organization (IMO) Marine Environment Protection Committee (MEPC) regulations have limited sulfur in marine fuels from 3.5% by mass to 0.5%, or required exhaust gas cleaning systems (scrubbers) to achieve an equivalent reduction in sulfur oxide ($SO_x$) pollution (IMO, 2019). These IMO 2020 fuel sulfur regulations and the resulting decrease in sulfate aerosol (airborne particulates) are expected to have large benefits to public health (Partanen et al., 2013; Sofiev et al., 2018; Zhang et al., 2021). They are also expected to have an undesired side effect, however: as sulfate aerosol cools the climate by reflecting sunlight directly and indirectly via changing cloud properties, the IMO 2020 $SO_x$ reductions may accelerate global warming.

Shipping effects on clouds were first identified in in the mid-1960s in satellite imagery of ship tracks, or curvilinear cloud perturbations following individual ships (Conover, 1966; Twomey et al., 1968). For the same amount of liquid water within a cloud, increasing aerosol increases the cloud droplet number concentration ($N_d$) and decreases the cloud-top effective radius ($r_e$), brightening the clouds (Twomey, 1974, 1977). Cloud macrophysical adjustments to this aforementioned Twomey

effect have been observed within ship tracks as well, and can reinforce the microphysical brightening effect by suppressing drizzle (Albrecht, 1989; Goren and Rosenfeld, 2012) or counteract it by enhancing entrainment (Chen et al., 2012; Coakley and Walsh, 2002; Toll et al., 2019). Understanding how much greenhouse gas warming is masked by these aerosol–cloud interactions from shipping and other forms of pollution is the largest source of uncertainty in quantifying present-day anthropogenic radiative forcing (Forster et al., 2021).

Although ship tracks have long served as "natural experiments" for testing hypotheses about aerosol–cloud interactions in cases of clear causality (Christensen et al., 2022), until recently, attempts to observationally assess regional-to-global-scale cloud perturbations and forcing from shipping have found negligible (Schreier et al., 2007) or null effects (Peters et al., 2011) due to the small fraction of ships that form easily identifiable tracks and the large background variability in cloud properties. New methods using machine learning have identified many times more ship tracks than has been possible with

manual identification (Watson-Parris et al., 2022; Yuan et al., 2022; Yuan et al., 2019) and analyses tracking air masses from ship locations have shown that cloud adjustments differ systematically between easily-identifiable and "invisible" ship tracks (Manshausen et al., 2022). Using some of these newer methods, it has been shown that ship track occurrence decreased regionally after the introduction of emission control areas around North America and Europe and then globally after the IMO 2020 regulations went into effect (Gryspeerdt et al., 2019; Watson-Parris et al., 2022; Yuan et al., 2022). Large-scale changes

in cloud microphysical and macrophysical properties have been more equivocal, however. Yuan et al. (2022) found smaller $N_d$ increases within ship tracks after the IMO 2020 regulations, as expected, but, paradoxically, greater $r_e$ decreases than before and no difference in cloud brightness. Watson-Parris et al. (2022) did not find evidence for a change in global or regional $N_d$ after the IMO 2020 regulations despite the clear decrease in ship tracks, with a possible exception in the southeastern Atlantic.

In this work, we assess the detectability of large-scale cloud perturbations from the IMO 2020 regulations by revisiting

an alternate solution to the limitations of "bottom-up" methods tracking individual ship tracks: a "top-down" statistical approach developed by Diamond et al. (2020), hereafter D20, to identify regional-scale cloud perturbations within a shipping corridor in the southeastern Atlantic Ocean basin. A unique meteorological setup makes that region ideal for estimating causal aerosol effects: near-surface winds blow parallel to the shipping corridor and closely constrain the pollution, which also happens to intersect a major stratocumulus cloud deck. D20 used a universal kriging method (see Zimmerman and Stein, 2010,

and references therein) to estimate counterfactual fields of cloud properties and radiation in the absence of the shipping corridor based on the observed spatial statistics of nearby, non-shipping-affected grid boxes. They found significant increases in $N_d$ and cloud albedo (a measure of cloud reflectivity) and decreases in $r_e$ within the stratocumulus deck but estimated that several years' worth of data was needed to detect a clear signal. Thus, it is possible that the effect of the IMO 2020 regulations will have just become detectable using their method.

Here, we apply an updated version of the D20 universal kriging algorithm to satellite retrievals of $r_e$ and overcast albedo ($A_{cld}$; top-of-atmosphere albedo when clouds are present) from the Clouds and the Earth's Radiant Energy System (CERES) Single Scanner Footprint (SSF) product for the Terra satellite (Loeb et al., 2018; Minnis et al., 2011). The reader is referred to Appendix A: Methods for further details about the data, universal kriging algorithm, and significance tests.

Although D20 found a substantial decrease in cloud liquid water path within the corridor during the afternoon, no significant
cloud macrophysical adjustments were found in the morning. We therefore interpret any changes in $r_e$ and $A_{cld}$ using the Terra
record (observations at ~10:30 AM local time) as being dominated by the Twomey effect. We focus on both the austral spring
season (SON; September–October–November), which features the strongest shipping signal [likely due to a combination of
favorable meteorology and lower background $N_d$ (Grosvenor et al., 2018)], and the annual mean (ANN), which averages a
greater number of observations and thus should minimize noise. For a variable $X$, the "factual" or observed value in the
presence of the shipping corridor is referred to as "Ship" ($X_{Ship}$), the counterfactual value in the absence of shipping obtained
via kriging is referred to as "NoShip" ($X_{NoShip}$), and the Ship-NoShip difference is signified as $\Delta X$ and is interpreted as the
effect due to the presence of the shipping corridor.

## 2 Results

An unambiguous decrease in the magnitude of the $r_e$ perturbation within the shipping corridor is evident in the post-regulation
(2020–2022) data as compared to the pre-2020 climatology (2002–2019) and the immediately preceding three-year period
(2017–2019) during austral spring (Fig. 1). Although several significant grid boxes (observations falling outside the 95%
confidence interval of the counterfactual) remain in the south of the domain, and thus some level of continued shipping
influence is detected (as indicated by field significance at the $\ll 0.05$ level), the microphysical changes are smaller and less
clearly tied to the corridor; the signal is completely lost further north. Similar results are found for the annual mean values
(Fig. S1), albeit with a clearer continued effect of shipping in the 2020–2022 data.

The shipping perturbation in overcast albedo is less well defined than that in the cloud microphysics, but there is still
a clear perturbation in the 2002–2019 climatology and 2017–2019 data that is diminished in the 2020–2022 data in austral
spring (Fig. 2). Similar results are found in the annual mean, although the 2020–2022 change is more ambiguous from visual
inspection alone (Fig. S2). Lower background $A_{cld}$ values in 2020–2022, particularly in the annual mean (Fig. S2g), may be
related to unusually warm sea surface temperatures (Figs. S3–4); as dimmer clouds are relatively more susceptible to aerosol
perturbations, this effect may partially obscure the decrease in cloud brightening from the IMO 2020 regulations.

To assess how anomalous the post-regulation 2020–2022 shipping perturbation values are, we compare them to those
from prior three–year periods by averaging over a core shipping corridor region (see Methods in Appendix A) and, to minimize
effects from changing background conditions, also calculate perturbations as relative differences ($100\% * \Delta X / X_{Ship}$). Full results
are reported in Table S1 and summarized in Fig. 3. For the austral spring $r_e$ perturbations, 2020–2022 is unprecedentedly weak
(Fig. 3a) and does not overlap any prior period's value within their 95% confidence intervals (Table S1). The separation
between 2020–2022 and any other period's values is not as clear for austral spring $A_{cld}$ (Fig. 3b), although the 2020–2022
perturbation values are the lowest on record and are the only period for which the effect is not distinguishable from zero at the
95% confidence level (Table S1). For the annual mean $r_e$ perturbations, the 2020–2022 values are lower than any other period
and the difference with the climatological value is much larger than for any other period, although the separation is not as clear

as for austral spring (Fig. 3c). While the annual mean $A_{cld}$ perturbations for 2020–2022 are also the lowest on record, the difference from climatology is not extreme compared to other periods (Fig. 3d).

To assess whether a reduction in the shipping effect after the IMO 2020 regulations went into effect is detected at various possible levels of confidence, Table S2 reports different percentiles of the ratio of the 2020–2022 relative differences over the climatology. A decrease in the $r_e$ perturbation is detected at greater than 99% confidence in the austral spring and at greater than 95% confidence in the annual mean, whereas decreases in the $A_{cld}$ perturbation are only significant at the 90% confidence level in the austral spring and within the interquartile range in the annual mean. We thus conclude that the effect of the IMO 2020 regulations has been clearly detected in the large-scale cloud microphysics and that there is strong evidence for a decrease in cloud brightness, although more years of data may be required for unequivocal detection of changes in overcast albedo.

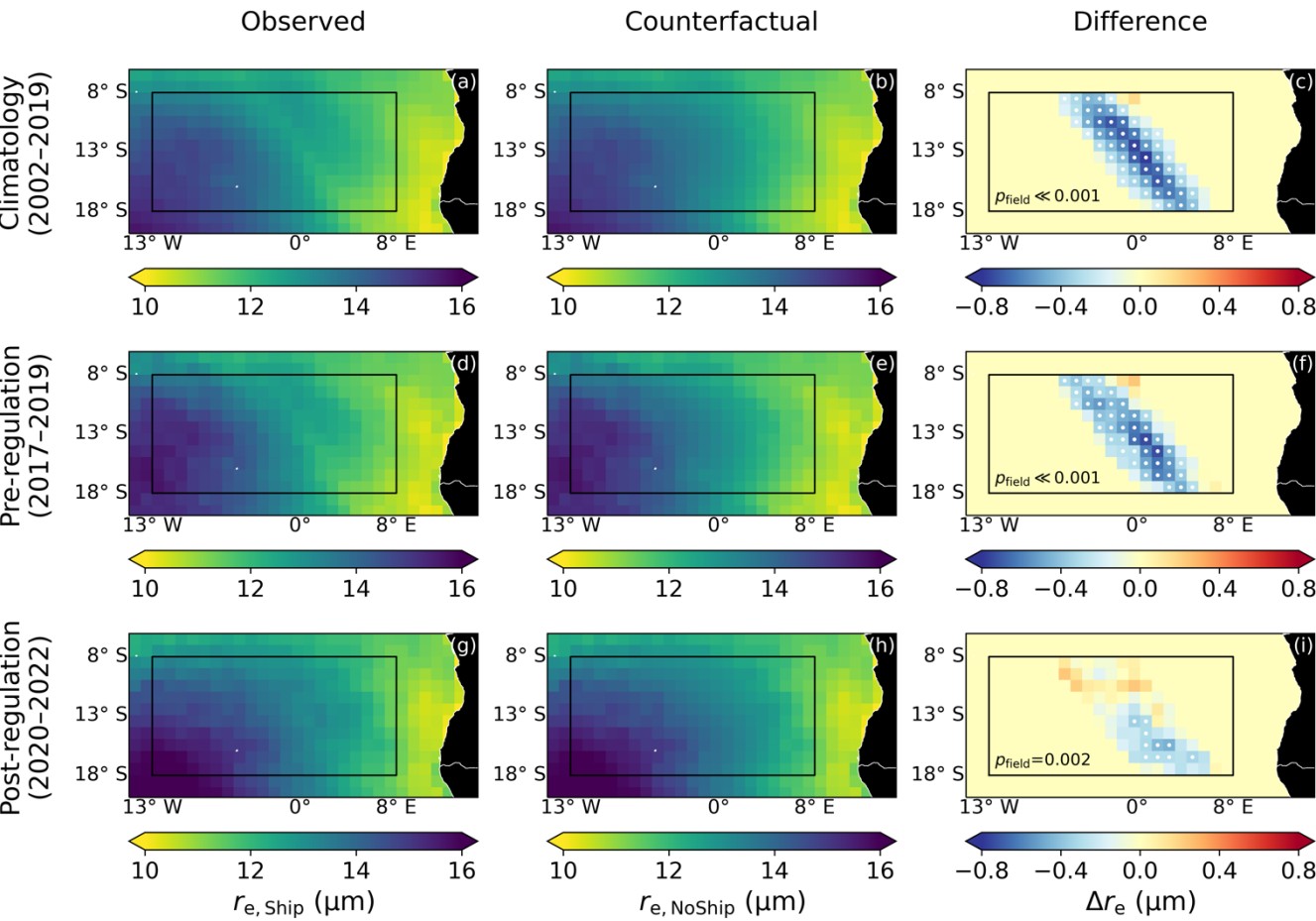

**Figure 1.** Maps of factual (observed) and counterfactual values and their difference for austral spring cloud-top effective radius for the pre-2020 climatology (a–c), the immediately pre-regulation 3-year period 2017–2019 (d–f), and the immediate post-regulation 3-year period 2020–2022 (g–i). The analysis domain of 18° S to 8° S, 13° W to 8° E is outlined in black. Grid points for which the observed values fall outside the 95% confidence interval obtained via kriging are indicated by white dots and the corresponding field significance values are reported in (c,f,i).

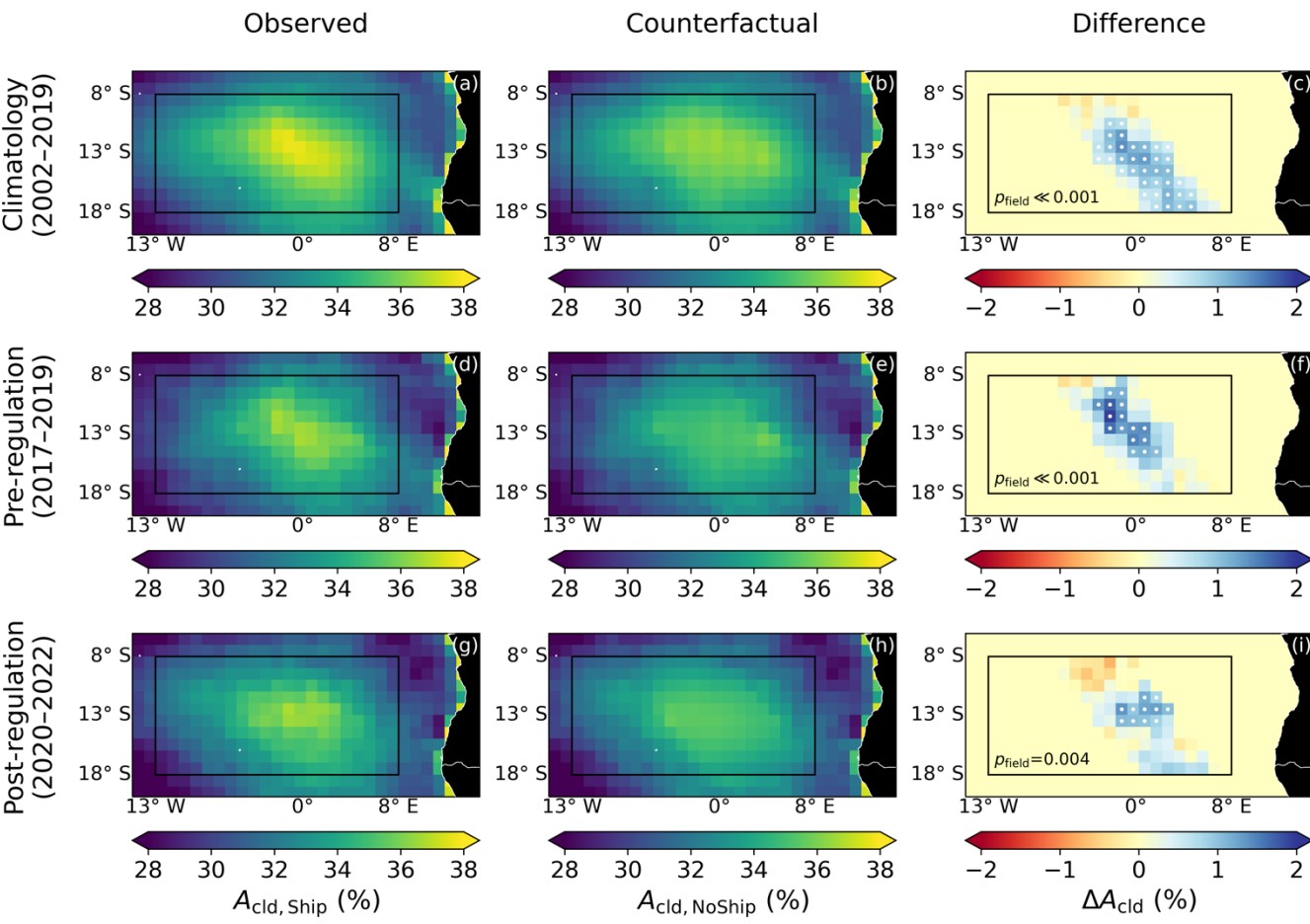

**Figure 2.** As in Fig. 1, but for austral spring overcast albedo.


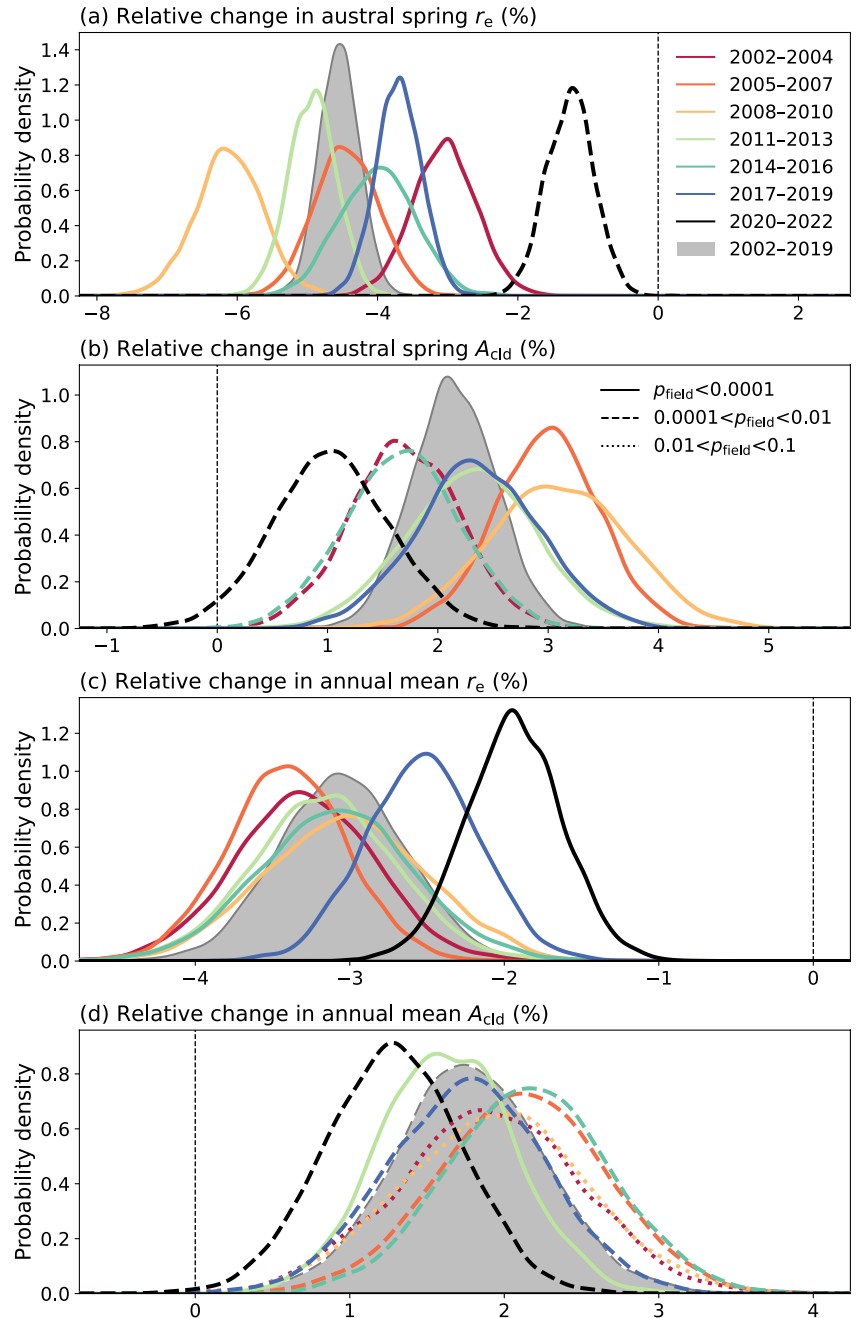

**Figure 3.** Probability densities (via Gaussian kernel density estimation) for the Ship-NoShip relative differences within the core shipping corridor for austral spring $r_e$ (a), austral spring $A_{cld}$ (b), annual mean $r_e$ (c), and annual mean $A_{cld}$ (d). The 2002–2019 climatology values are shown as gray shading, the three-year periods prior to the IMO 2020 regulations as colored lines, and the 2020–2022 period as black lines. Solid, dashed, and dotted lines indicate decreasing degrees of field significance.

# 3 Discussion

## 3.1 Monitoring compliance with IMO regulations

Assessing (non)compliance with the IMO 2020 regulations is of critical importance for ensuring that the intended public health benefits are realized. One assessment method is to monitor the sulfur content of the global fuel oil supply. According to data supplied to the IMO MEPC (IMO, 2020, 2021, 2022, 2023), before 2020, the average sulfur mass content of marine fuel oils was ~2.5% and ~80% of the global fuel oil supply exceeded 0.5%; since 2020, the average sulfur mass content declined to ~1% and only ~20% of fuel exceeds 0.5% (Fig. S5). These values understate compliance, as a "carriage ban" forbids ships

from carrying the remaining noncompliant fuel oil unless they have scrubbers installed (IMO, 2018). Geophysical monitoring via cloud changes, as has been shown in Yuan et al. (2022) and Watson-Parris et al. (2022) for ship track occurrence and here for large-scale cloud microphysical properties, offers an independent check to increase confidence that there has been substantial compliance with the IMO 2020 regulations. As our improving of the cloud effects from shipping aerosol improves, it may become possible to assess regional differences in compliance or even compliance for individual ships, complementing

other successful geophysical monitoring programs like those for detecting ozone depleting substances (Montzka et al., 2018; Park et al., 2021; Rigby et al., 2019).

Given the clear detection of cloud microphysical changes in austral spring after the IMO 2020 regulations went into effect, it is reasonable to ask whether advanced statistical methods are necessary for evaluating (some level of) compliance or if simple time series [e.g., Fig. S5 of Watson-Parris et al. (2022)] would suffice. From the time series of austral spring Ship

and NoShip $r_e$ values averaged over the southeastern Atlantic (Fig. 4), it is evident that the shipping effect before 2020–2022 is of similar magnitude to interannual variability in the background values and that the 2020–2022 $r_e$ values are estimated to be the highest on record even before any IMO 2020 effect is considered. As an estimate of what the 2020–2022 observed value would have been under a scenario of complete noncompliance with the sulfur regulations, the average Ship-NoShip difference from the 2002–2019 climatology is applied to the 2020–2022 NoShip value (Noncompliance in Fig. 4). The +0.1 μm difference

between the observed (Ship) value and this noncompliance hypothetical is due to compliance with the IMO 2020 regulations. If we had rather based our noncompliance scenario on a persistence forecast of the 2017–2019 value and then observed the value from the "true" noncompliance estimate calculated above, we would erroneously conclude that the IMO 2020 regulations were successfully implemented and led to a +0.3 μm increase in regional $r_e$. Of course, in this latter scenario, the true value of the difference due to IMO 2020 would have been zero and the apparent effect only an artifact of the changing background.

Caution is therefore advised in attempting to interpret time series of large-scale cloud properties without applying a method (like track identification or kriging) that plausibly establishes causality.

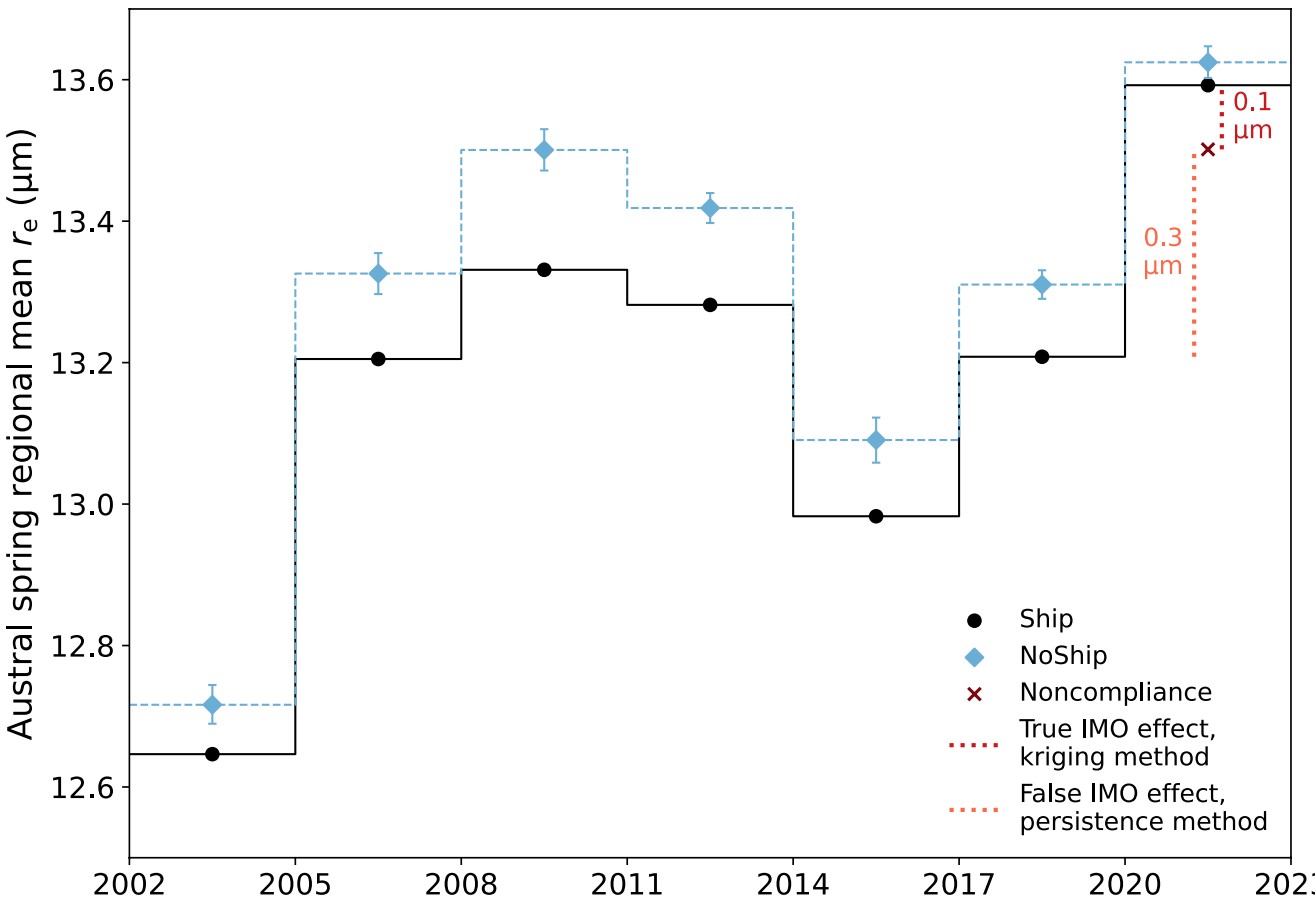

**Figure 4.** Time series of observed Ship (black circles) and mean NoShip (blue diamonds) values averaged the southeastern Atlantic analysis domain (18° S to 8° S, 13° W to 8° E) for austral spring $r_e$. Error bars represent 95% confidence for the NoShip values. A Noncompliance scenario in which the IMO 2020 regulations were not enforced and the Ship-NoShip difference in 2020–2022 were the same as for the 2002–2019 climatology is denoted as a dark red "x". The red dotted line denotes the estimated effect from compliance with the IMO 2020 regulations, calculated as the difference between the observed Ship value and the hypothetical Noncompliance value expected for no change in 2020–2022. The orange dotted line denotes the mistakenly determined effect that would have resulted if the Noncompliance scenario were true and observed but a persistence forecast of 2017–2019 were used as the expectation value for no change in 2020–2022.

## 3.2 Radiative forcing implications

Assuming that the Terra-based $r_e$ and $A_{cld}$ perturbations are dominated by the Twomey effect as in D20, it is possible to estimate the instantaneous radiative forcing due to aerosol–cloud interactions (IRF$_{ACI}$; Forster et al., 2021) from the IMO 2020 regulations within the shipping corridor (see Methods in Appendix A). Results are shown in Fig. 5 for the 2002–2019 climatology, 2020–2022, and their difference (interpreted as the effect of the IMO 2020 regulations). The Twomey effect estimates are much better constrained for the calculations using $r_e$, but those using $A_{cld}$ show consistent results. The IMO 2020 regulations led to an ~2 W m$^{-2}$ IRF$_{ACI}$ within the shipping corridor during austral spring and an ~0.5 W m$^{-2}$ IRF$_{ACI}$ in the annual mean. Applying this ~35-70% decline in IRF$_{ACI}$ to the -0.1 to -0.6 W m$^{-2}$ range of forcing due to shipping emissions from climate models (Capaldo et al., 1999; Lauer et al., 2007; Peters et al., 2013; Righi et al., 2011; Sofiev et al., 2018), global forcing values of $O$(0.1 W m$^{-2}$) due to the IMO 2020 regulations are plausible. The strongest shipping effect in Lauer et al. (2007) represented 40% of their global ACI; a 70% reduction from that fraction would represent a forcing of $0.2 \pm 0.1$ W m$^{-2}$ based on the currently assessed IRF$_{ACI}$ value of $0.7 \pm 0.5$ W m$^{-2}$, or $0.3 \pm 0.2$ W m$^{-2}$ including adjustments (Forster et al., 2021).

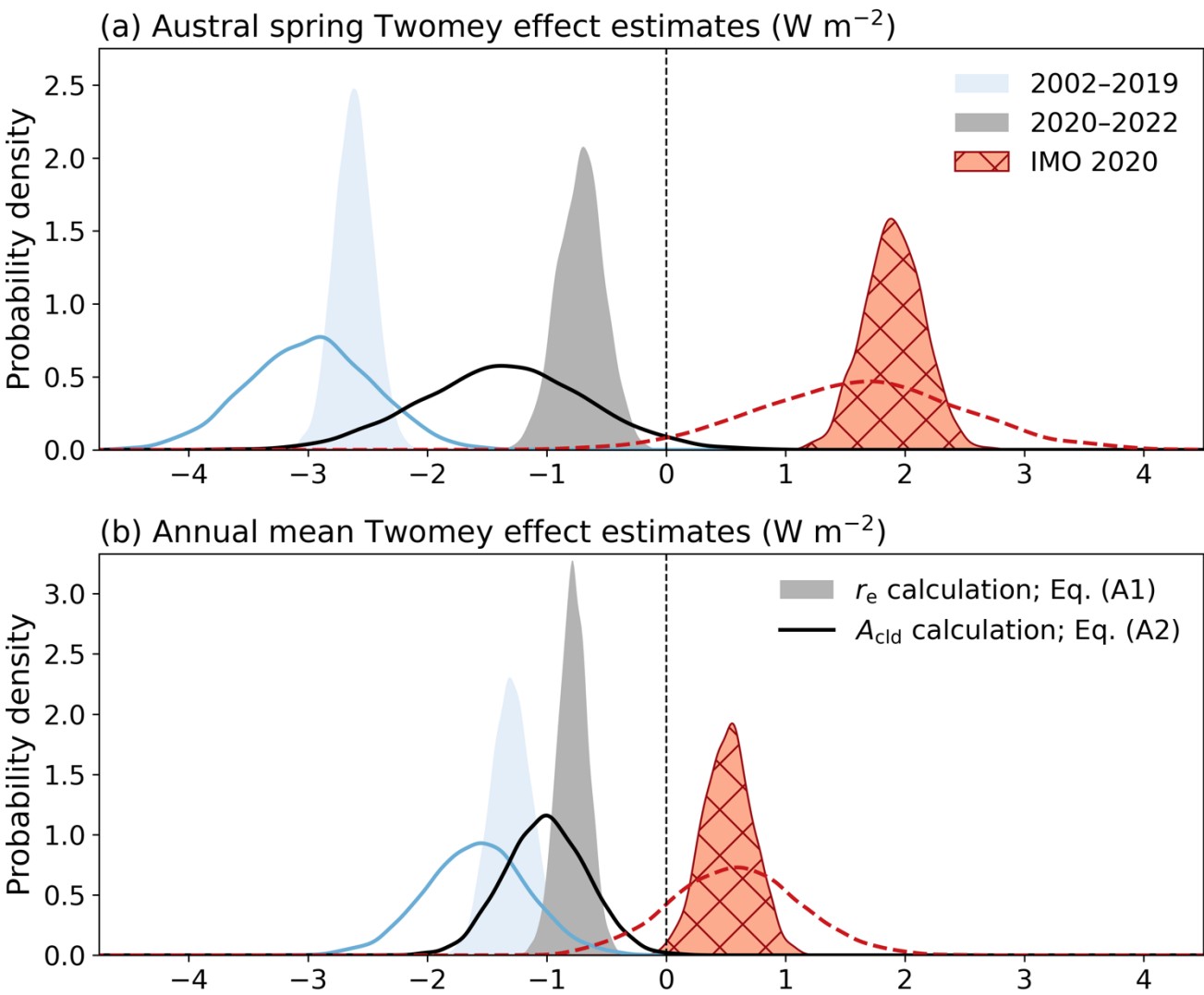

**Figure 5.** Probability densities (via Gaussian kernel density estimation) of $\text{IRF}_{\text{ACI}}$ for austral spring (a) and the annual mean (b) over the core shipping corridor calculated using the changes in $r_e$ (shading) from Eq. (A1) and $A_{\text{cld}}$ (lines) from Eq. (A2) for the 2002–2019 pre-regulation climatology (light solid blues) and 2020–2022 post-regulation period (dark solid grays) due to the presence of the shipping corridor and as the 2020–2022 minus climatology difference as an estimate of the effect due to implementation of the IMO 2020 regulations (patterned reds).

185

## 4 Conclusions

There is a detectable change in large-scale cloud microphysical properties and evidence supporting a decrease in cloud brightening within the major southeastern Atlantic shipping corridor after implementation of the IMO 2020 fuel sulfur regulations, resulting in a positive $IRF_{ACI}$ within the corridor of $O(1 \text{ W m}^{-2})$. Although this study did not address potential changes in cloud adjustments from the IMO 2020 regulations, this will be an important area of future work, especially as the fuel regulations are expected not only to decrease overall aerosol numbers but also shift them toward smaller sizes and sootier composition (Ault et al., 2010; Lack et al., 2011; Seppälä et al., 2021).

## Appendix A: Methods

### Data

All cloud, radiation, and meteorological data in this work come from the CERES SSF regional 1° x 1° (SSF1deg) monthly product based on the CERES instrument from the Terra satellite (CERES Science Team, 2021, 2023; Loeb et al., 2018; Wielicki et al., 1996). Radiative fluxes are temporally interpolated over the diurnal cycle assuming constant cloud and meteorological properties but varying the solar zenith angle (Doelling et al., 2013); our results therefore reflect the diurnal average assuming constant Terra conditions rather than the instantaneous midmorning value, which would be much greater in magnitude, but do not account for any diurnal cloud evolution. Overcast albedo values are calculated as in D20 but with the clear-sky albedo assumed to be 0.1 to avoid issues with missing clear-sky data in the SSF1deg product. The constant clear-sky albedo may cause a high bias in the absolute $A_{cld}$ values, especially during the southern African biomass burning season (June to October), but this effect should be small given the very overcast conditions and would not strongly affect the observed versus counterfactual differences. The overcast albedo (albedo as seen from space when clouds are present) differs from the cloud albedo (cloud reflectivity) due to the scattering and absorption of sunlight from above-cloud aerosols and gases.

Cloud properties are retrieved from Moderate Resolution Imaging Spectroradiometer (MODIS) measurements using CERES algorithms (CERES Science Team, 2016; Minnis et al., 2011), which have some differences from the standard MODIS products (Platnick et al., 2017). Only daytime cloud retrievals utilizing 3.7 μm channel radiances are used in this work. Low cloud fraction is defined for clouds with cloud top effective pressure values greater than 700 hPa.

Meteorological variables including surface skin temperature (over oceans, the Reynold's sea surface temperature), estimated inversion strength (Wood and Bretherton, 2006), and wind speed are from the NASA Goddard Space Flight Center Global Modeling and Assimilation Office (GMAO) Goddard Earth Observing System (GEOS) version 5.4.1 (CERES Science Team, 2021).

Sulfur dioxide ($SO_2$) emissions data from 2010 are from the Emissions Database for Global Atmospheric Research (EDGAR) version 4 (Crippa et al., 2018) and are identical to those used in D20. The EDGAR $SO_2$ values are only used for identification of the shipping corridor location.

**Shipping corridor identification**

For each latitude between 8° S and 18° S, shipping-affected grid boxes are identified as those with the maximum EDGAR $SO_2$ emission values between 13° W and 8° E as well as the four grid boxes to the west and two to the east. This represents a northward and westward expansion of the shipping corridor definition used in D20 for their subtropical domain and is intended to better center the microphysical effects. Ship tracking via the automatic identification system (AIS) identifies substantial traffic slightly west of where EDGAR places the maximum $SO_2$ emissions and there are indications of an additional westward shift in traffic during 2020 (March et al., 2021). As a sensitivity test, the analysis in Fig. 1 was repeated using a shipping corridor mask shifted further west by two degrees, but no notable differences were found. The core shipping corridor area used in Figs. 3 and 5 and Tables S1-2 is defined as the central three grid boxes of the shipping mask for each latitude.

**Universal kriging**

The universal kriging algorithm mostly follows the implementation of D20, using the geoR statistical package (Ribeiro and Diggle, 2018). Universal kriging is a classic geostatistical method (Zimmerman and Stein, 2010) that has been widely employed in the geosciences and other fields (Chilès and Desassis, 2018), in which estimates of unknown values at some location are informed by nearby observations of the same variable under the assumption that errors around a mean function are spatially correlated as a function of the distance between locations only (stationarity). In our case, counterfactual values for the shipping-affected grid boxes identified above are estimated using the values of nearby, non-shipping-affected grid boxes between 8° S and 18° S and 13° W and 8° E. Our mean function takes the form of a multiple linear regression model using as regressors some combination of the surface skin temperature (SST), estimated inversion strength (EIS), and wind speed (WS) from the SSF1deg auxiliary data and latitude (lat), longitude (lon), and their squares ($lat^2$, $lon^2$) and product (lat*lon), as determined by whichever combination minimizes the Bayesian information criterion (BIC) to avoid overfitting. Table S1 reports the selected combination of regressors (based on BIC minimization) for each combination of variable ($r_e$ and $A_{cld}$) and time period. A logit transform is applied to the $A_{cld}$ values before kriging, which was found by D20 to produce more normally distributed errors around the mean function for bounded fields like albedo and cloud fraction. The stationary error term is then estimated by using weighted least squares to fit a parametric (exponential) covariance model to an empirical variogram (a plot of the squared difference between pairs of variables versus their distance). Figures S6-9 show the binned empirical variograms and fitted variograms (see Zimmerman and Stein, 2010) for austral spring $r_e$ and logit($A_{cld}$) and annual mean $r_e$ and logit($A_{cld}$), respectively. Using the statistical model provided by the kriging process above (Ribeiro and Diggle, 2018), we simulate 5,000 realizations of the NoShip counterfactual for each variable/time period combination.

**Statistical significance testing**

250  Four distinct tests of statistical significance are used in this work, the first three following D20. Statistical significance for individual shipping-affected grid boxes is assessed as whether the observed Ship value exceeds the 97.5[th] percentile or falls below the 2.5[th] percentile of the distribution obtained via kriging for the counterfactual NoShip value for that grid box.

Field significance is assessed by determining whether the number of individually significant grid boxes calculated above is extreme as compared to that which could occur by chance under the null hypothesis that the region is unaffected by 255  shipping; p-values ($p_{field}$) are calculated as the fraction of the 5,000 NoShip simulations that would have a number of individually significant grid boxes greater than or equal to the factual case and are adjusted for multiple testing using a Benjamini-Hochberg adjustment to control the false discovery rate (Benjamini and Hochberg, 1995; Ventura et al., 2004). If none of the 5,000 NoShip simulations would produce a number of individually significant grid boxes as or more extreme than the Ship field, $p_{field}$ is reported as <0.0001 instead of zero in Table S1. All $r_e$ perturbations (except 2020–2020 austral spring) 260  are field significant at a <0.0001 level; the $A_{cld}$ perturbations have more variation, although all are significant at greater than 90% confidence (Fig. 3 and Table S1). Interpreting the field significance as a measure of the robustness of the shipping effects, we should therefore have greatest confidence in the $r_e$ results and least (but still a good deal of) confidence in the annual $A_{cld}$ results.

The range of Ship-NoShip values generated from the 5,000 simulated NoShip fields is used to assess if the magnitude 265  of effects within the core shipping corridor area are statistically distinct from zero at 95% confidence (Table S1).

Finally, a new test for "detectability" at different confidence interval thresholds is presented in Table S2 and based on the range of possible ratios of 2020–2022 over climatological relative Ship-NoShip values from the 5,000 simulated NoShip fields. We adopt significance at 95% confidence or greater as distinguishing between "detection" for the $r_e$ changes versus "evidence" short of detection for the $A_{cld}$ changes.

270  **Twomey effect calculations**

For the $r_e$ perturbations, $IRF_{ACI}$ is estimated following Eq. (A1):

$$IRF_{ACI} = -F_{\odot} C_{low} \phi_{atm} \alpha_{cld} (1 - \alpha_{cld}) (-\Delta r_e / r_{e,Ship}), \tag{A1}$$

where $F_{\odot}$ is the insolation, $C_{low}$ is the low cloud fraction, $\phi_{atm}$ is a transfer function between changes in overcast and cloud albedo (Diamond et al., 2020; Wood, 2021), and $\alpha_{cld}$ is the cloud albedo. Based on the values in D20, $\phi_{atm}$ is estimated as 0.6 275  and $\alpha_{cld}$ as 0.5.

For the $A_{cld}$ perturbations, $IRF_{ACI}$ is estimated following Eq. (A2):

$$IRF_{ACI} = -F_{\odot} C_{low} \Delta A_{cld}. \tag{A2}$$

Eqs. (A1) and (A2) neglect liquid water path and cloud fraction adjustments to the Twomey effect. The effective radiative forcing due to aerosol-cloud interactions ($ERF_{ACI}$), accounting for cloud adjustments, would be greater in magnitude 280  than calculated here if cloudiness were increased via drizzle suppression and lesser if cloudiness were decreased via enhanced

entrainment. D20 found that adjustments were small in the morning but substantially offset brightening during the afternoon in austral spring. The apparently small effects in the morning may reflect diurnal competition between precipitation suppression, which maximizes overnight, and entrainment drying, which maximizes during the day (Sandu et al., 2008). Thus, the IRF$_{ACI}$ values here are likely larger than ERF$_{ACI}$ values would be after accounting for adjustments over the full diurnal cycle, at least in austral spring.

**Code availability**

Code for processing the data and recreating the analyses in this work is available from GitHub (https://github.com/michael-s-diamond/IMO2020, last accessed 12 June 2023). The universal kriging algorithm is implemented in R (R Core Team, 2014) using the geoR package (Ribeiro and Diggle, 2018). Other analyses are performed in Python using the numpy (Harris et al., 2020), cartopy (Met Office, 2010-2015), matplotlib (Hunter, 2007), scipy (Virtanen et al., 2020), statsmodels (https://github.com/statsmodels/statsmodels, last accessed 10 May 2023) and xarray (Hoyer and Hamman, 2017) packages.

**Data availability**

SSF1deg data (CERES Science Team, 2023) are available from the NASA Langley Research Center CERES ordering tool (https://ceres.larc.nasa.gov/data/, last accessed 14 April 2023). EDGAR data (European Commission Joint Research Centre, 2018) are available from the European Commission Joint Research Centre Data Catalogue (https://doi.org/10.2904/JRC_DATASET_EDGAR, last accessed 10 May 2023). Processed data used in this work (Diamond, 2023) are available in a Zenodo repository (https://doi.org/10.5281/zenodo.7864530, last accessed 12 June 2023).

**Competing interests**

The author declares that they have no conflict of interest.

**Acknowledgements**

This work was supported with startup funds from Florida State University. Hannah M. Director merits continued gratitude for her contributions to the original code base and analysis methods. Discussion with Leon Simons provided the impetus for attempting to detect a signal with only three years of post-2020 data. Thanks to Clare Singer, Emily de Jong, and an anonymous reviewer for their constructive comments.

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
