# Peer review of "Detection of large-scale cloud microphysical changes within a major shipping corridor after implementation of the IMO 2020 fuel sulfur regulations"

_EGUsphere, 2023_

## Referee Comment (RC1)

**Review of "Detection of large-scale cloud microphysical changes and evidence for decreasing cloud brightness within a major shipping corridor after implementation of the International Maritime Organization 2020 fuel sulfur regulations" by Diamond (egusphere-2023-971)**

The study analyzes cloud microphysical changes in the southeast Atlantic using satellite data and advanced statistical methods. The author utilizes a recent regulation on sulfur in shipping fuel to constrain how aerosol-cloud interactions affect the radiative properties of clouds by comparing data from before and after the regulation's implementation. The author shows, inter alia, that the regulation causes an increase in the cloud droplet effective radius and a decrease in cloud albedo in the years after the regulation, an expected result due to the reduced number of sulfate aerosol particles produced by the ships. Overall, this is a relevant study, well-written, and I only have a few minor suggestions. Thus, I fully support the manuscript's publication in ACP Letters.

**Minor Comments**

Ll. 29 ff.: I believe the author refers to the cloud-top effective radius in the remainder of the study.

Ll. 51 – 53: I understand the benefits of the chosen region to analyze the effects of the regulation. However, can such an analysis be conducted for other parts of the globe? What differences are expected?

Ll. 80 – 82: The author writes that the effect of the regulation on the annually averaged cloud albedo is more ambiguous. The author indicates that this is due to the lower background cloud albedo. However, a lower background cloud albedo should be more susceptible to changes in the aerosol or cloud droplet concentration, and thus provide a stronger signal. Please elaborate on this.

Ll. 159 – 161: Considering that the multi-year cloud-top effective radius reaches values of up to 13.6 µm (Fig. 4), it is likely that (some) clouds produce drizzle. Thus, liquid water and cloud fraction adjustments will accompany the Twomey effect. How would they affect the forcing derived in ll. 163 –168?

**Technical Comments**

Ll. 63 ff.: While the abbreviation D20 has been introduced in l. 52, its use is somewhat erratic.

Figs. 1 and 2: I suggest increasing the font of the panel labels.

Fig. 2: The center column shows $A_{cld,NoShip}$, not $A_{cld,Ship}$. Adapt the title of the contour label bar.

Fig. 5b: I recommend replacing the equations with something more accessible.

---

## Referee Comment (RC2)

**Review of "Detection of large-scale cloud microphysical changes and evidence for decreasing cloud brightness within a major shipping corridor after implementation of the International Maritime Organization 2020 fuel sulfur regulations" by Diamond**

This study looks at how cloud microphysical properties (effective radius and brightness) have responded to the 2020 IMO regulation of sulfur in shipping fuels. They examine changes in the southeast Atlantic shipping corridor using statistical methods to construct counterfactual fields of cloud properties without shipping aerosols. They show that only 3 years of data are necessary to already see robust changes in effective radius and cloud albedo due to the sulfur regulation, despite large interannual variability.

Overall, this is a very interesting study. The paper is well written and clear. In general, I felt that the discussion of methodology and results could be expanded a bit to make the paper more readable, especially for those unfamiliar with the previous Diamond et al. (2020) work. I have the following comments, which I think would improve this manuscript.

**Major comments:**
- It would be nice to include more information about the methods in the main text rather than relegating it to the appendix. I found it necessary to read the methodology first in many cases to understand the figures and key points of the manuscript. In particular, the sections on "Universal kriging" and "Statistical significance testing" would be most beneficial to include before the presentation of results.
- Fig 4:
  o A more substantive question that arose from this figure: Why are data only reported from non-overlapping 3-year time windows? Could, for example, the analysis be done and this figure be made showing a 3-year running-mean over the time period? How would that change the calculation of IMO effect via the persistence method?
  o Can error bars be added to this figure to show an estimate of the confidence that the NoShip effective radii are in fact larger.
  o A technical point: The bracket on 0.3 um should extend the full height of the dotted line.

**Minor comments:**
- I recommend making a shorter, more direct title. Maybe just remove the phrase "and evidence for decreasing cloud brightness within a major shipping corridor"
- L9: add "*may* come with an undesired" because this is the question the paper sets out to prove or disprove
- L33-34: Delete "Challenges in" & change "pollution are" to "pollution *is*"
- L46: change to "Yuan et al. (2022) found *smaller* Nd *increases*" to parallel the phrasing of "greater re decreases"
- L67: Add a sentence here explaining the choice of season, or why SON features the strongest shipping signal. I assume it is because Sc are most prominent during this season, so there is more baseline cloud which has the potential to be brightened, but this would be helpful to make explicit.

- Fig 1/2: More descriptive labels on the figures would be helpful. E.g. instead of just labeling the years, add "Pre-regulation climatology (2002-2019)", "3 years pre-regulation (2017-2019)", "3 years post-regulation (2019-2022)". And instead of using the ambiguous names "Ship" and "NoShip" these columns could be labeled as "Measurements" and "Inferred Counterfactual"
- Fig 3: This figure is great, and very rich. It warrants more than 1 paragraph of discussion in the text. In particular, it would be nice to include some more detail on how pfield is calculated and then interpretation of what the pfield values mean. Is it significant that the 2020-2022 years are the only 3-year mean that has a change in re with pfield > 0.0001?
- L120: It would be interesting to put this section on compliance into more context in the geophysical literature. This is not the first time that geophysical data have been useful in assessing compliance with policy regulations (e.g. remote sensing monitoring of CFCs and methane leakage from oil and gas). How does your work fit into that bigger picture?
- L164: Does the difference between 2 W/m2 and 0.5 W/m2 in the seasonal vs annual mean give an estimate of how the cloud susceptibility to aerosols varies seasonally? This could be an interesting idea to pursue quantitatively in the context of MCB.
- L167: Can you put this estimate of 0.4 W/m2 into more context? First, what is the baseline value (the total IRF_ACI) this shipping term is modifying? Second, how does this compare to the IRF_ARI from shipping?
- Fig 5: 1) Put "Twomey effect (W m-2)" as the x-label rather than in the subplot titles. 2) Either define the mathematical expressions in the legend in the caption, or (even better) change the legend labels to something more interpretable, 3) consider using more B/W-friendly colors for this plot.
- L192: Please elaborate on the assumptions made for these data products. What bias does assuming the constant cloud and meteorological properties over the diurnal cycle introduce?
- L194: Assuming a constant clear-sky albedo of 0.1 seems like it would ignore the presence of aerosols (dust or smoke). Is this a problem for this region? How much does this bias your results?
- L209: It could be helpful to include a map of the EDGAR SO2 overlaid with AIS ship tracks to illustrate the discussion of the section on "Shipping corridor identification"
- L219: Please add some references on the kriging algorithm, for its development, and also a bit of discussion for how this algorithm is used by others in the literature. Just from reading this section is sounds as if Diamond et al. (2020) was the first/only study to use this method, but of course, this is a fairly common geostatistical technique.
- L226: What is the physical reasoning for including lat^2, lon^2, and lat*lon as predictors in the multiple linear regression?
- L232: This one sentence is the only mention of Figures S5-S8. These figures are not explained anywhere in the supplement and difficult to interpret alone. If they are not going to be discussed in the paper, I would suggest removing them entirely because they do not aid understanding without much more explanation about what an empirical variogram is, how it is computed, and what they show us about the kriging method.

*This paper was reviewed by Clare E. Singer, including discussion with Emily K. de Jong.*

---

## Author Comment (AC1)

Thanks to the anonymous reviewer for their positive assessments of the manuscript and helpful suggestions for further improvement. Please find detailed responses below in blue. -MD

**Reviewer 1**

*Ll. 29 ff.: I believe the author refers to the cloud-top effective radius in the remainder of the study.*

Yes; "cloud-top" has now been added here and incorporated into the $r_e$ abbreviation.

*Ll. 51 – 53: I understand the benefits of the chosen region to analyze the effects of the regulation. However, can such an analysis be conducted for other parts of the globe? What differences are expected?*

Unfortunately, I have not yet identified another shipping corridor that seems promising for this method. There are a few corridors between Hawaii and California that could be a good combined target, but the perturbation is much more diffuse and thus the ability to estimate a believable counterfactual with nearby, non-shipping-affected regions is limited. However, different methods that could skillfully predict cloud properties for a given meteorological state and aerosol background and then evaluate changes for a large aerosol excursion while controlling for any coincident meteorological changes (e.g., Y. Chen et al, 2022; Wall et al., 2022) could be useful for this problem (as this is the subject of in-progress work, I will refrain from speculating much further here!). I would expect perturbations in shallow stratocumulus clouds under strong inversions to be similar to those in the southeast Atlantic and perturbations in trade cumulus regions to be weaker due to a combination of weaker boundary layer coupling and potentially diminished microphysical susceptibility in that regime. A new project starting this fall will examine meteorological controls on the cloud responses in the southeast Atlantic in more detail, which will be useful for extrapolation to other regions and regimes.

The uniqueness of the southeast Atlantic region is now better emphasized in the text: "A unique meteorological setup makes that region ideal for estimating causal aerosol effects: near-surface winds blow parallel to the shipping corridor and closely constrain the pollution, which also happens to intersect a major stratocumulus cloud deck."

Chen, Y., Haywood, J., Wang, Y., Malavelle, F., Jordan, G., Partridge, D., et al. (2022). Machine learning reveals climate forcing from aerosols is dominated by increased cloud cover. *Nature Geoscience, 15*, 609–614.

Wall, C. J., Norris, J. R., Possner, A., McCoy, D. T., McCoy, I. L., & Lutsko, N. J. (2022). Assessing effective radiative forcing from aerosol-cloud interactions over the global ocean. *Proc Natl Acad Sci U S A, 119*(46), e2210481119.

*Ll. 80 – 82: The author writes that the effect of the regulation on the annually averaged cloud albedo is more ambiguous. The author indicates that this is due to the lower background cloud albedo. However, a lower background cloud albedo should be more susceptible to changes in the aerosol or cloud droplet concentration, and thus provide a stronger signal. Please elaborate on this.*

Thank you for raising this point; the original phrasing was unclear. The point has been elaborated: "Lower background $A_{cld}$ values in 2020–2022, particularly in the annual mean (Fig. S2g), may be related to unusually warm sea surface temperatures (Figs. S3–4); as dimmer clouds are relatively more susceptible to aerosol perturbations, this effect may partially obscure the decrease in cloud brightening from the IMO 2020 regulations."

*Ll. 159 – 161: Considering that the multi-year cloud-top effective radius reaches values of up to 13.6 µm (Fig. 4), it is likely that (some) clouds produce drizzle. Thus, liquid water and cloud fraction adjustments will accompany the Twomey effect. How would they affect the forcing derived in ll. 163 –168?*

Drizzle maximizes in the southeast Atlantic during September, which may help explain the lower background $N_d$. However, in D20, we found that adjustments tended to offset the Twomey effect overall because of decreased liquid water path from enhanced entrainment (with statistically insignificant cloud fraction changes), at least in the afternoon. The lack of strong adjustments in the morning may reflect diurnal cancellation between enhanced cloudiness from precipitation suppression overnight and depleted cloudiness from entrainment effects during the day (e.g., Sandu et al., 2008).

A discussion has been added in the methods after describing how the Twomey effect alone is calculated: "Eqs. (A1) and (A2) neglect liquid water path and cloud fraction adjustments to the Twomey effect. The effective radiative forcing due to aerosol-cloud interactions (ERF$_{ACI}$), accounting for cloud adjustments, would be greater in magnitude than calculated here if cloudiness were increased via drizzle suppression and lesser if cloudiness were decreased via enhanced entrainment. D20 found that adjustments were small in the morning but substantially offset brightening during the afternoon in

austral spring. The apparently small effects in the morning may reflect diurnal competition between precipitation suppression, which maximizes overnight, and entrainment drying, which maximizes during the day (Sandu et al., 2008). Thus, the $IRF_{ACI}$ values here are likely larger than $ERF_{ACI}$ values would be after accounting for adjustments over the full diurnal cycle, at least in austral spring."

Sandu, I., Brenguier, J.-L., Geoffroy, O., Thouron, O., & Masson, V. (2008). Aerosol Impacts on the Diurnal Cycle of Marine Stratocumulus. *Journal of the Atmospheric Sciences, 65*(8), 2705-2718.

*Ll. 63 ff.: While the abbreviation D20 has been introduced in l. 52, its use is somewhat erratic.*

Fixed.

Figs. 1 and 2: I suggest increasing the font of the panel labels.

Done.

*Fig. 2: The center column shows Acld,NoShip, not Acld,Ship. Adapt the title of the contour label bar.*

Thank you for catching this typo; fixed.

*Fig. 5b: I recommend replacing the equations with something more accessible.*

Done.

---

## Author Comment (AC2)

Thanks to Clare Singer and Emily de Jong for their positive assessments of the manuscript and helpful suggestions for further improvement. Please find detailed responses below in blue. -MD

**Reviewer 2 (Clare Singer with input from Emily de Jong)**

*It would be nice to include more information about the methods in the main text rather than relegating it to the appendix. I found it necessary to read the methodology first in many cases to understand the figures and key points of the manuscript. In particular, the sections on "Universal kriging" and "Statistical significance testing" would be most beneficial to include before the presentation of results.*

Unfortunately, the ACP Letters format precludes accepting the reviewer's suggestion in totality. The separate Methods section within an Appendix is a fundamental difference between the new ACP Letters article type (with a strict word limit in the main text) and a more traditional, longer-form ACP article. As all the necessary methodological information to interpret the results is included in the Appendix, and readers are referred to my earlier paper establishing the method for this application for additional details, I believe the manuscript is appropriate for the ACP Letters format.

That said, I agree that it would be helpful to include as much relevant information for interpretation nearby the results as practicable and to explicitly refer the reader to the Appendix for specific methodological information. The following lines have been added:

Introduction and approach: "The reader is referred to Appendix A: Methods for further details about the data, universal kriging algorithm, and significance tests."

Results: "Although several significant grid boxes (observations falling outside the 95% confidence interval of the counterfactual) remain in the south of the domain, and thus some level of continued shipping influence is detected (as indicated by field significance at the $\ll 0.05$ level), the microphysical changes are smaller and less clearly tied to the corridor; the signal is completely lost further north."

*Fig 4: A more substantive question that arose from this figure: Why are data only reported from non-overlapping 3-year time windows? Could, for example, the analysis be done and this figure be made showing a 3-year running-mean over the time period? How would that change the calculation of IMO effect via the persistence method?*

This is a good question and interesting idea. There are two main reasons not to extend the analysis to all possible consecutive 3-year periods. The first has to do with our application of a Benjamini-Hochberg false detection rate correction for multiple tests — running a larger number of tests means a larger risk of false positives, so the correction penalizes all p-values based on the total number of tests. The second is that performing that many tests would be labor intensive — although the kriging algorithm is mostly automated, it is very sensitive to the initial choice of fitting parameters for the semivariogram and can easily fail to converge on reasonable values, so the user must visually inspect the variogram and ensure reasonable values are selected. As including non-independent 3-year periods would provide minimal new information while degrading confidence across the board, it would be hard to justify the effort.

The persistence method calculations are meant to be more illustrative, showing that plausible choices under that method could result in very misleading values. This overall conclusion would be very unlikely to change with better time resolution or accounting for trends.

*Fig 4: Can error bars be added to this figure to show an estimate of the confidence that the NoShip effective radii are in fact larger.*

Done.

*Fig 4: A technical point: The bracket on 0.3 um should extend the full height of the dotted line.*

Brackets have been removed.

*I recommend making a shorter, more direct title. Maybe just remove the phrase "and evidence for decreasing cloud brightness within a major shipping corridor"*

Thanks for the suggestion. The title has been abbreviated to: "Detection of large-scale cloud microphysical changes within a major shipping corridor after implementation of the IMO 2020 fuel sulfur regulations".

*L9: add "may come with an undesired" because this is the question the paper sets out to prove or disprove*

Added.

*L33-34: Delete "Challenges in" & change "pollution are" to "pollution is"*

Changed.

*L46: change to "Yuan et al. (2022) found smaller Nd increases" to parallel the phrasing of "greater re decreases"*

Changed.

*L67: Add a sentence here explaining the choice of season, or why SON features the strongest shipping signal. I assume it is because Sc are most prominent during this season, so there is more baseline cloud which has the potential to be brightened, but this would be helpful to make explicit.*

The seasonality of the shipping signal is a matter of ongoing work. There are a few different factors explaining why the perturbation may maximize in SON, and I do not yet have a good quantification of their relative contributions. Sc being most prominent in this season (with cloud fraction approaching 100% in the center of the domain) is certainly a contributing factor. At least in MERRA-2, the $SO_4$ perturbation maximizes in SON, which could be related to in-cloud chemical processing, although it could also be related to the strength of the constraining along-corridor winds. The coupling state of the boundary layer would also matter for transport to cloud base. The southeast Atlantic has an unusual seasonal cycle in $N_d$ compared to the other major Sc decks (e.g., Fig. 9 in Grosvenor et al., 2018), with a peak in July followed by a minimum in September, so the background $N_d$ is also likely a major factor.

A parenthetical note has been added: "We focus on both the austral spring season (SON; September–October–November), which features the strongest shipping signal [likely due to a combination of favorable meteorology and lower background $N_d$ (Grosvenor et al., 2018)], and the annual mean (ANN), which averages a greater number of observations and thus should minimize noise."

Grosvenor, D. P., Sourdeval, O., Zuidema, P., Ackerman, A., Alexandrov, M. D., Bennartz, R., et al. (2018). Remote Sensing of Droplet Number Concentration in Warm Clouds: A Review of the Current State of Knowledge and Perspectives. *Reviews of Geophysics, 56*, 409–453.

*Fig 1/2: More descriptive labels on the figures would be helpful. E.g. instead of just labeling the years, add "Pre-regulation climatology (2002-2019)", "3 years pre-*

*regulation (2017-2019)", "3 years post-regulation (2019-2022)". And instead of using the ambiguous names "Ship" and "NoShip" these columns could be labeled as "Measurements" and "Inferred Counterfactual"*

For Figures 1-2 and S1-2, the y-labels have been changed to "Climatology (2002–2019)", "Pre-regulation (2017–2019)", and "Post-regulation (2020–2022)" and the x-labels to "Observed", "Counterfactual", and "Difference".

*Fig 3: This figure is great, and very rich. It warrants more than 1 paragraph of discussion in the text. In particular, it would be nice to include some more detail on how pfield is calculated and then interpretation of what the pfield values mean. Is it significant that the 2020-2022 years are the only 3-year mean that has a change in re with pfield > 0.0001?*

Thanks for the positive words about the Figure. Unfortunately, due to the space constraints of the ACP Letters format, it does not make sense to expand the discussion of Figure 3 in the main text, as something else would need to be cut. The following additional information has been added to the description of field significance in the Methods: "All $r_e$ perturbations (except 2020–2020 austral spring) are field significant at a <0.0001 level; the $A_{cld}$ perturbations have more variation, although all are significant at greater than 90% confidence (Fig. 3 and Table S1). Interpreting the field significance as a measure of the robustness of the shipping effects, we should therefore have greatest confidence in the $r_e$ results and least (but still a good deal of) confidence in the annual $A_{cld}$ results." It has also been clarified that the purpose of the field significance test is to try and reject the "null hypothesis that the region is unaffected by shipping".

The importance of 2020–2022 being the only period with $p_{field}$ > 0.0001 (for austral spring $r_e$) is a matter of interpretation. I generally prefer to view p-values as indications of confidence/strength of evidence rather than strict cutoffs. The higher p-value and weaker effect size in 2020–2022 both point to the effect of the IMO 2020 regulations reducing the strength of the shipping perturbation. However, as mentioned in the text, the 2020–2022 $p_{field}$ of 0.002 is still highly significant by any reasonable measure and thus is evidence for some level of continuing, albeit weaker, shipping effects.

*L120: It would be interesting to put this section on compliance into more context in the geophysical literature. This is not the first time that geophysical data have been useful in assessing compliance with policy regulations (e.g. remote sensing monitoring of CFCs and methane leakage from oil and gas). How does your work fit into that bigger picture?*

Thanks for the great idea! I've added references for the CFC-11 saga (detection of global increase, pinpointing of China, confirmation of solution) as an example of a successful monitoring regime: "As our improving of the cloud effects from shipping aerosol improves, it may become possible to assess regional differences in compliance or even compliance for individual ships, complementing other successful geophysical monitoring programs like those for detecting ozone depleting substances (Montzka et al., 2018; Park et al., 2021; Rigby et al., 2019)."

Based in part on external feedback, I've also added a new Figure S5 showing the IMO MEPC fuel oil statistics for reference and slightly expanded their discussion: "According to data supplied to the IMO MEPC (IMO, 2020, 2021, 2022, 2023), before 2020, the average sulfur mass content of marine fuel oils was ~2.5% and ~80% of the global fuel oil supply exceeded 0.5%; since 2020, the average sulfur mass content declined to ~1% and only ~20% of fuel exceeds 0.5% (Fig. S5)."

Montzka, S. A., Dutton, G. S., Yu, P., Ray, E., Portmann, R. W., Daniel, J. S., et al. (2018). An unexpected and persistent increase in global emissions of ozone-depleting CFC-11. *Nature, 557*(7705), 413-417.

Rigby, M., Park, S., Saito, T., Western, L. M., Redington, A. L., Fang, X., et al. (2019). Increase in CFC-11 emissions from eastern China based on atmospheric observations. *Nature, 569*(7757), 546-550.

Park, S., Western, L. M., Saito, T., Redington, A. L., Henne, S., Fang, X., et al. (2021). A decline in emissions of CFC-11 and related chemicals from eastern China. *Nature, 590*(7846), 433-437.

*L164: Does the difference between 2 W/m2 and 0.5 W/m2 in the seasonal vs annual mean give an estimate of how the cloud susceptibility to aerosols varies seasonally? This could be an interesting idea to pursue quantitatively in the context of MCB.*

Agreed that this is an interesting angle! I have a project starting this fall to drill down into the sub-seasonal meteorological variability in addition to the seasonality discussed above, which I think will be highly relevant for the question of susceptibility for MCB. The SON versus annual estimates here are affected by both the change in the size of the effect seasonally (smaller in the other months than SON) and the lower cloud fraction annually (75%) versus in SON (~90%). Adjustments, which are not addressed in this work, are likely to play a large role in addition to the Twomey effect.

*L167: Can you put this estimate of 0.4 W/m2 into more context? First, what is the baseline value (the total IRF_ACI) this shipping term is modifying? Second, how does this compare to the IRF_ARI from shipping?*

Thanks for the request here. In thinking about it more, I decided to adjust the calculation a bit from a simple application of the 70% figure to the upper Lauer et al. (2007) estimate of -0.6 W $m^{-2}$ (~0.4 W $m^{-2}$), as their global ERF$_{ACI}$ in that scenario is a very strong -1.5 W $m^{-2}$. Instead, I'm now using the latest IPCC figures and estimating an upper(ish) bound by applying the 70% reduction to their 40% shipping contribution estimate: "Applying this ~35-70% decline in IRF$_{ACI}$ to the -0.1 to -0.6 W $m^{-2}$ range of forcing due to shipping emissions from global models (Capaldo et al., 1999; Lauer et al., 2007; Peters et al., 2013; Righi et al., 2011; Sofiev et al., 2018), global forcing values of O(0.1 W $m^{-2}$) due to the IMO 2020 regulations are plausible. The strongest shipping effect in Lauer et al. (2007) represented 40% of their global ACI; a 70% reduction from that fraction would represent a forcing of 0.2 ± 0.1 W $m^{-2}$ based on the currently assessed IRF$_{ACI}$ value of 0.7 ± 0.5 W $m^{-2}$, or 0.3 ± 0.2 W $m^{-2}$ including adjustments (Forster et al., 2021)."

Ongoing work using a different method will hopefully provide a better-constrained estimate of the global ERF$_{ACI}$ from the IMO 2020 regulations, although the rough calculation here is useful in thinking about the plausible magnitude range.

Shipping IRF$_{ARI}$ is negligible compared to the IRF$_{ACI}$ — for example, Lauer et al. (2007) calculated a global mean direct forcing of -0.01 W $m^{-2}$ for the same case with an indirect forcing of -0.60 W $m^{-2}$. The emitted particles tend to be Aitken mode, so there isn't much effective surface area even for a decent number concentration.

*Fig 5: 1) Put "Twomey effect (W m-2)" as the x-label rather than in the subplot titles. 2) Either define the mathematical expressions in the legend in the caption, or (even better) change the legend labels to something more interpretable, 3) consider using more B/W- friendly colors for this plot.*

For 1), the current format matches Fig. 3 and is more compact. 2) The legend has been simplified. 3) I had tried getting different shades for B/W friendliness, but the differences are probably too subtle. The easiest solution is to add a pattern to the IMO 2020 estimate and keep the climatology as solid/light and 2020–2020 as solid/dark.

*L192: Please elaborate on the assumptions made for these data products. What bias does assuming the constant cloud and meteorological properties over the diurnal cycle introduce?*

I'm not sure I would characterize this as a bias per se — the CERES SSF processing gives fluxes that represent an equivalent diurnal forcing, versus the much larger (for Terra morning overpass time) instantaneous forcing one would get from a product like the hourly resolved CERES SYN. Of course, this equivalent diurnal forcing does not actually account for the diurnal cycle of cloudiness or potential changes in cloud adjustments over the course of the day. This has now been clarified in the text: "Radiative fluxes are temporally interpolated over the diurnal cycle assuming constant cloud and meteorological properties but varying the solar zenith angle (Doelling et al., 2013); our results therefore reflect the diurnal average assuming constant Terra conditions rather than the instantaneous midmorning value, which would be much greater in magnitude, but do not account for any diurnal cloud evolution."

*L194: Assuming a constant clear-sky albedo of 0.1 seems like it would ignore the presence of aerosols (dust or smoke). Is this a problem for this region? How much does this bias your results?*

This is a good point, as large quantities of smoke are present over the southeast Atlantic from June-October. Given the high cloud fraction, the practical effect is probably very small, however; the overcast albedo is sensitive to $(1-C)A_{clr}/C$. Since the seasonal smoke plume should have a smooth spatial gradient without any discontinuities over the shipping corridor, a retrieval bias wouldn't necessarily be that problematic for the observed minus counterfactual difference even if it biases the absolute value of each. For that reason, I thought the constant background clear-sky albedo assumption was better than introducing a clear-sky albedo stitched together from multiple different time periods or from a different CERES product (like EBAF), although the practical of any of those options again should be small.

This has now been addressed in the methods: "The constant clear-sky albedo may cause a high bias in the absolute $A_{cld}$ values, especially during the southern African biomass burning season (June to October), but this effect should be small given the very overcast conditions and would not strongly affect the observed versus counterfactual differences."

*L209: It could be helpful to include a map of the EDGAR SO2 overlaid with AIS ship tracks to illustrate the discussion of the section on "Shipping corridor identification"*

This would be a nice plot! Unfortunately, the AIS ship data is very expensive to access and has onerous restrictions on being shared. March et al. (2021) were allowed to share their data on 2020 anomalies in ship traffic (not the absolute values), but this wouldn't be as useful to plot.

*L219: Please add some references on the kriging algorithm, for its development, and also a bit of discussion for how this algorithm is used by others in the literature. Just from reading this section is sounds as if Diamond et al. (2020) was the first/only study to use this method, but of course, this is a fairly common geostatistical technique.*

Thanks for this comment; I agree that much of this was simply taken for granted in the original presentation. The reference to the geoR statistical package is now included in the methods in addition to the code availability statement upon first mention of the algorithm and again for the simulation method (which geoR includes in a convenient routine, whereas some of the Python I looked at briefly may not include). Although geoR is used here, there are a number of packages in R and Python (and I am sure other languages) that could be used as well.

In lieu of picking a few specific applications at random (e.g., statistical downscaling of precipitation data), I have now cited a useful retrospective of kriging algorithms in the geosciences (and other) literature: "Universal kriging is a classic geostatistical method (Zimmerman and Stein, 2010) that has been widely employed in the geosciences and other fields (Chilès and Desassis, 2018), in which…". I have also added a note hinting at the broader literature to the first introduction of kriging in Section 1: "D20 used a universal kriging method (see Zimmerman and Stein, 2010, and references therein)". The Zimmerman and Stein chapter in Gelfand et al.'s *Handbook of Spatial Statistics* has been my main reference in developing the code and interpreting results; I'm sure there are equally good chapters in any textbook on geostatistics, but I'm not sure how helpful it would be to readers to list more here.

Chilès, J.-P., & Desassis, N. (2018). Fifty Years of Kriging. In *Handbook of Mathematical Geosciences* (pp. 589-612).

*L226: What is the physical reasoning for including lat^2, lon^2, and lat\*lon as predictors in the multiple linear regression?*

There isn't really any difference in reasoning between the linear lat and lon regressors and their squares and product — the fundamental goal of the mean function in our

case is simply to capture the spatial trend in the non-shipping-affected grid boxes. The regression coefficients do not necessarily hold any physical significance, including those for the cloud controlling factors in our analysis. This is in contrast to a method like that used in Wall et al. (2022), which develops a regression model in order to interpret the coefficients. Our insight is that we do not need to know the exact relationships between the meteorology and background aerosol in the non-shipping-affected regions to have some degree of confidence that the spatial gradients should be relatively smooth over a sufficiently long averaging period in the absence of a localized perturbation (like the shipping corridor). In this sense, the current method is analogous to inpainting for a generative AI like DALL-E 2. If we did have a method that could highly accurately predict cloud and radiative properties based on meteorology and background aerosol alone at a relatively fine resolution (1 degree or better), that could replace the mean function (or potentially the kriging algorithm entirely in favor of a machine learning method that could learn a nonstationary correlated spatial error term) — this is the subject of ongoing work and the new project starting this fall.

Wall, C. J., Norris, J. R., Possner, A., McCoy, D. T., McCoy, I. L., & Lutsko, N. J. (2022). Assessing effective radiative forcing from aerosol-cloud interactions over the global ocean. *Proc Natl Acad Sci U S A, 119*(46), e2210481119.

*L232: This one sentence is the only mention of Figures S5-S8. These figures are not explained anywhere in the supplement and difficult to interpret alone. If they are not going to be discussed in the paper, I would suggest removing them entirely because they do not aid understanding without much more explanation about what an empirical variogram is, how it is computed, and what they show us about the kriging method.*

The variograms are included for completeness, transparency, and reproducibility; the essential feature of the figures is that the fit is not obviously horrible/violating the assumptions of kriging. They would potentially be useful to someone trying to replicate the results from scratch using the code on my GitHub. The fitting parameters (including initial guesses, which the algorithm can be sensitive to) and binned semivariances are already included in the Zenodo data, but it would be a more work to create the plots versus just reference them in the SI. 99% or more of readers can safely disregard, but I do think they are potentially useful for the rare case. The readers are now referred again to the Zimmerman and Stein (2010) chapter as a useful resource: "Figures S6-9 show the binned empirical variograms and fitted variograms (see Zimmerman and Stein, 2010)".